# FIFA football nurse – A task sharing approach in sports and exercise medicine practice in grassroots women's football in low- and middle- income settings. A study protocol for a cluster randomised controlled trial

Nonhlanhla Sharon Mkumbuzi[1,2,3,4‡]*, Andrew Massey[5], Samuel Kiwanuka Lubega[6], Ben Sorowen[7], Enock Madalitso Chisati[8‡]

1 Department of Sports, Exercise, and Rehabilitation, Northumbria University, Newcastle upon Tyne, United Kingdom, 2 Department of Rehabilitation, Midlands State University, Gweru, Zimbabwe, 3 Department of Human Movement Science, Nelson Mandela University, Gqeberha, South Africa, 4 NtombiSport (PTY) Ltd. Cape Town, South Africa, 5 FIFA Medical Department, FIFA, Zurich, Switzerland, 6 Department of Sports Science, Kyambogo University, Kampala, Uganda, 7 Department of Mathematics and Statistics, Kyambogo University, Kampala, Uganda, 8 Department of Rehabilitation Sciences, Kamuzu University of Health Sciences (KUHeS), Blantyre, Malawi

‡ NSM and EMC are joint senior authors on this work.
* nonhlanhla.mkumbuzi@northumbria.ac.uk

## Abstract

Football (soccer) is a very popular team sport among African women and girls, with player numbers continuing to rise at all levels of the sport. Whereas the participation in football and associated injuries are on the rise, there are not enough sports and exercise medicine (SEM) personnel to attend to these women football players. While Africa may not currently have enough SEM trained medical doctors and/or physiotherapists, it has relatively higher numbers of other healthcare workers; for example, nurses, who lead healthcare services provision from community to tertiary levels. The primary objective of this study will be to compare sports medicine practices; injury prevention behaviours; injury risk parameters; incidence and prevalence of injuries and illnesses in teams with and without a Football Nurse during one competitive season in Malawi's Women's football league. This study will be a cluster randomised control trial will recruit 24 teams from the Women's Football League in Malawi, which will be randomised to either the intervention group or the control group. A cohort of 12 nurses will receive training in basic football medicine; after which they will be attached to a total of 12 women's football teams (intervention group) during one competitive season. The Football Nurses will be directly report to a physiotherapist or doctor in their district to whom they will refer serious injuries for investigations, or further management. The teams with Football Nurses will be compared to other teams that will not have Football Nurses. We expect to develop a low cost, sustainable and context relevant solution to manage the treatment gap of football injuries/illnesses in underserved communities such as women's football.

**Data Availability Statement:** Deidentified research data will be made publicly available upon request when the study is completed and published.

**Funding:** The present study is funded by the Women's Football Development Department at Federation Internationale Football Association (FIFA). The funders had no role in study design, data collection and analysis, decision to publish, or preparation of the manuscript.

**Competing interests:** AM is Director of the Medical Department at FIFA; NSM is a technical consultant for FIFA's Women's Football Department. This does not alter our adherence to PLOS ONE policies on sharing data and materials

**Trial registration number:** Pan African Clinical Trial Registry (PACTR202205481965514).

# Introduction

## Women's football

Football (soccer) is a very popular sport among women and girls in Africa and the numbers continue to rise at all levels of the sport [1, 2]. The world football governing body has shown its commitment to improving the women's game by providing $ 1 billion in funding to the Women's Football Development Department [3]. Similarly, national associations are also investing more in their women's teams and more commercial sponsors such as celebrities are coming on board to sponsor women's football [4]. Further, the recently concluded WEURO and AWCON Championships broke attendance and viewership records [5, 6], which shows the growing interest in women's football as a spectator sport.

This increase in participation and professionalisation of the women's game is also directly associated with an increase in football related injuries [7–9]. Typical football injuries/adverse events include ligament sprains, muscle strains, or concussions [8, 10]. However, while the tide in women's football is turning towards a more professional approach in the Global North, the same cannot be said about progress in the Global South or at grassroots level, where resources, human or otherwise, continue to be scarce. Hence, while the physiology of football injuries may be the same, injuries and illnesses in women's football players in low- and middle-income countries (LMICs) such as in sub-Saharan Africa (SSA) have different socioeconomic implications for management and return to play to those faced by their counterparts in high income countries (HIC). This is, in part, because trained medical care is very limited in the former [11–14] compared to the latter [15].

Recent studies on African sport have shown that more than 50% of athletes do not have medical attendees at training or competition [12, 13, 16]. These findings may be attributed to the shortage of trained sports and exercise medicine (SEM) practitioners such as medical doctors (hereinafter referred to as doctors) and/or physiotherapists (PTs) in SSA. In such an environment of scarcity, it is the women, children and those with disabilities who bear the brunt of these shortages [17] and SEM is no different: medical care is very limited for women's, youth, and para football teams worldwide [18]. It is widely believed that available and suitably trained SEM personnel would sooner practice with a (adult) men's team where the benefits and conditions of service are more lucrative.

When women's teams do have medical cover, it tends to be variable with no continuity of practice and care, and players may need to pay for themselves [18]. Consequently, women players may suffer worse outcomes from injury because they are at the periphery of health services provision in football [19], which compromises their safe participation in the sport [20, 21]. SEM is a relatively new discipline in Africa, therefore the number of specialists is proportionately less than other established disciplines of medicine [12, 14]. Hence, while participation in football and injuries arising thereof are increasing, there are not enough SEM specialists to attend to these football players, which results in an enormous treatment gap in sports injuries. Consequently, 70% of injured athletes do not receive medical treatment or specialist referral following injury on the field of play [11].

## Nurses as health care providers in LMICs

A solution to this shortage of trained SEM professionals and subsequent treatment gap in women's football, proposed by the investigators, is to train more professionals. However, therein lies another challenge: there are currently not enough institutions that train doctors and/or PTs on the African continent to meet the basic health needs of their populations, let alone 'spare' for sport (**Table 1**).The current medical best practice in football requires to have at least a doctor, PT and conditioning coach at the pitch side during training and matches [21–23]. This best practice model is based on personnel availability in HIC (**Table 1**). Therefore, it may seem impossible to achieve in most teams in LMICs. As a result, most football teams in Africa, especially women's teams, have to do without medical care at training or practice except perhaps at national team level.

While SSA may not currently have enough doctors and PTs, SSA also tends to have relatively higher numbers of other health workers; namely, nurses [25]. For example, Zimbabwe has 0.24 PTs; 1.6 doctors and 7.2 nurses to 10 000 citizens [26, 27] and close to 80% of all healthcare workers in South Africa are nurses [25]. Consequently, the bedrock of health systems in SSA is nurse-led. This heavy reliance on nurses as the main healthcare providers is reflected in the number of nursing schools and graduates in SSA. Zimbabwe and Uganda have 27 [28] and 50 institutions *(Lubega, 2021; personal communication)*, respectively, that train nurses while Malawi graduates approximately 2000 nurses annually *(Phiri, 2021; personal communication)*.

In the pyramidal referral system used in many LMICs, doctors are available from secondary health care (e.g. district hospital) and PTs are often available from tertiary health care (e.g. provincial hospital) while nurses are available from the community and primary care levels and throughout the referral pyramid [25, 29]. Hence, while access to a doctor or PT may be aspirational for most citizens/athletes of SSA, discretionary even, nurses are available to provide healthcare from grassroots level and sometimes healthcare facilities are exclusively staffed by nurses [25, 30]. A mirror strategy of having nurses available to provide health care from the community and primary care levels with doctors and PTs available at higher levels may be used in football medicine to ensure that health care services are provided for all participants at all levels of the game, especially at grassroots.

## Inclusion of nurses in task sharing in SEM

Perhaps, instead of having an SEM strategy that is heavily reliant on doctors and PTs as in HIC, we ought to adapt current global standards to fit the human resources profile of SSA and include the largest group of healthcare providers by numbers, nurses. In general medical practice, nurses currently conduct screening, diagnosis and management of medical conditions as well as referring to other healthcare workers as necessary and are an integral part of healthcare delivery in most LMICs [25, 30]. Similarly, they should be a key component to delivering health care services in football especially at grassroots levels and for underserved football players. Including nurses in the SEM practice agenda may help alleviate the shortage of pitch side medical care in women's football teams. Additionally, as nurses are predominantly women [31], their inclusion into SEM practice will also help increase representation of African women SEM practitioners as they are still underrepresented in the profession [32].

Therefore, the aim of this project is to develop a task sharing approach (Football Nurse) to actively recruit and train nurses as pitch side responders in grassroots women's football. Similar task sharing approaches have been used successfully in other sporting codes [33] and in LMICs to manage the diagnosis, treatment, and referral of other health conditions where there is a shortage of specialists [30]. The broad aim of this project is to explore the influence of

**Table 1. The number of institutions that train medical doctors and physiotherapists in three selected low-income African countries (Zimbabwe, Uganda, and Malawi) compared to those trained in a high-income country (Australia).**

| Country | Population (million) | Doctor training institutions (approx. number of graduates per year) | PT training institutions (approx. number of graduates per year) |
|---|---|---|---|
| **Zimbabwe** | 14.7 | 3 (200)[a] | 1 (25)[a] |
| **Uganda** | 44.3 | 6 (400)[b] | 2 (20)[b] |
| **Malawi** | 18.6 | 1 (80)[c] | 1 (25)[c] |
| **Australia** | 25.7 | 22 (3600) [24] | 24 (2500)[d] |

[a]Chibhabha, 2021

[b]Lubega, 2021

[c]Phiri, 2021

[d]Kemp, 2021

trained nurses as pitch side responders in grassroots women's football teams. Therefore, to minimise contamination, the unit of randomisation and analysis will be the football team.

## Rationale for components of the task sharing approach

Globally, women and girls in sports are confronted with the lack of gender friendly sports facilities, equipment, and support personnel [11]. While these barriers to safe participation of women in sport are widely acknowledged, there is a dearth of follow up interventions to specifically address these barriers. Despite an increase in participation in football, most women's football teams in Africa do not have medical care at training or during competition [11]. This is likely because there are not enough doctors and/or PTs to service these teams. The problem with the status quo is that this inadequate access to sports medical services increases the risk of injury, worsening of injury once incurred, or reinjury [13, 20, 21]. Hence, it is imperative that this gap be filled to ensure safe participation of women players in the game.

Nurses are the largest number of healthcare workers by numbers and are the backbone of most health systems in LMICs. Actively including nurses into SEM practice in these settings may provide an effective, affordable, and sustainable solution to bridge the treatment gap that African women football players currently face and allow safer participation in football activities for all, at all levels. The goal of the FIFA Football Nurse project will be to use a task sharing approach to train nurses to provide on field assessment and management of sports injuries in underserved communities such as grassroots women's football. This will ensure that the SEM needs of women football players are met from grassroots level.

## Research aim and hypotheses

The overall objective of the FIFA Football Nurse project will be to train nurses as first responders in grassroots women's football. The main aim of the project is to use the rigour of a randomised control trial (RCT) to evaluate the influence of trained nurses as first responders on sports injury related outcomes following Football Nurse intervention compared to usual treatment in grassroots women's football teams. The main outcomes will be at level of the team SEM practice (women's football teams are the clusters). We hypothesise that the FIFA Football Nurse intervention will be superior to the current management in grassroots women's football teams. Lastly, the project will evaluate the feasibility of training nurses as first responders in grassroots women's football teams.

### Specific objectives

#### Primary objective.

- To estimate the effect of football nurse intervention on team sports medicine practice during one competitive season in Malawi's Women's football league.

#### Secondary objectives.

- Determine current sports medicine practices in women's football teams in Malawi's Women's Football League.

- Determine baseline knowledge, attitudes, and behaviours of football stakeholders (players, coaches, team management) in women's football towards injury prevention programs in teams with and without a Football Nurse during one competitive season in Malawi's Women's football league.

- Identify football injury risk factors among women's football players in teams with and without a Football Nurse during one competitive season in Malawi's Women's football league.

- Determine the incidence and prevalence of injuries, illnesses, and medication use in teams with and without a Football Nurse during one competitive season in Malawi's Women's football league.

- Determine treatment and referral protocols of injuries and illnesses in teams with and without a Football Nurse during one competitive season in Malawi's Women's football league.

- To determine the financial feasibility of the FIFA Football Nurse intervention in grassroots women's football in Malawi.

We hypothesise that compared to teams without a Football Nurse, the mean sports medicine practice will be higher among teams with a Football Nurse during one competitive season in Malawi's Women's football league.

## Materials and methods

### Football nurse intervention

The purpose of this study will be to evaluate the effect on teams' sports medicine practice of randomising women's football teams to a trained general nurse as a pitch side responder or usual practice. This study will be a cluster randomised control trial, whose reporting will follow the Consolidated Standards of Reporting Trials (CONSORT) 2010 statement: extension to cluster randomised trials [34]. Clusters will be registered teams in the Women's Football League (24 teams) in Malawi. A 1x1 factorial design will be used to assess the effect of the FIFA Football Nurse intervention. The two allocated groups will be the football teams' (clusters) sport medicine practice. The teams will be randomly assigned to intervention or control. Prior to the commencement of the intervention, a survey will be sent out to the women's football teams to explore the current state of healthcare provision and policies for management or referral of players with injuries or illnesses, knowledge of injury prevention programs as well as their present medical needs. Following this, a cohort of 12 general nurses will receive a one (1) day training in basic football medicine using the latest version of the FIFA Emergency Care training manual. This training will be conducted by experienced sports medicine personnel as well as experienced trainers in basic life support (BLS) and first aid from the Departments of Emergency and Orthopaedics at Kamuzu University of Health Science (KUHeS), Blantyre,

Malawi. Following the training, the nurses will be attached to a total of 12 women's football teams during one competitive season. They will also be given access to reference materials that are adapted from the FIFA Emergency Care manual.

The roles of the nurses within the teams will include menstrual, basic health, injury risk assessment, nutritional, and mental health assessment; on field assessment, basic life support; immediate care, and referral of injuries; as well as maintaining records of injuries, illnesses, medication use and football exposure using standardised instruments as previously used in other studies [35, 36].

The trained nurses will be directly supervised by and report to a physiotherapist or doctor within their district (**Fig 1**) to whom they will refer players for intermediate care [11], refer serious injuries, refer players for investigations, or further management. The selected teams and nurses will be followed up for one season and outcomes measured at specified time points during the season as shown in **Fig 1**. Additionally, at the end of the season, semi- structured interviews will be conducted with the a selected sample of women football players and coaches from both arms of the study, as well as all nurses. The aim of these interviews will be to conduct an in- depth exploration of the participants' lived experiences and perceptions of the football nurse task sharing approach.

Current team medical practice; knowledge and practice of injury prevention strategies, injury risk; and mental health assessment will be assessed at the beginning, halfway through and at the end of the season. Football exposure; injuries, illnesses, and medication use surveillance will be recorded after every training session or match event throughout the season.

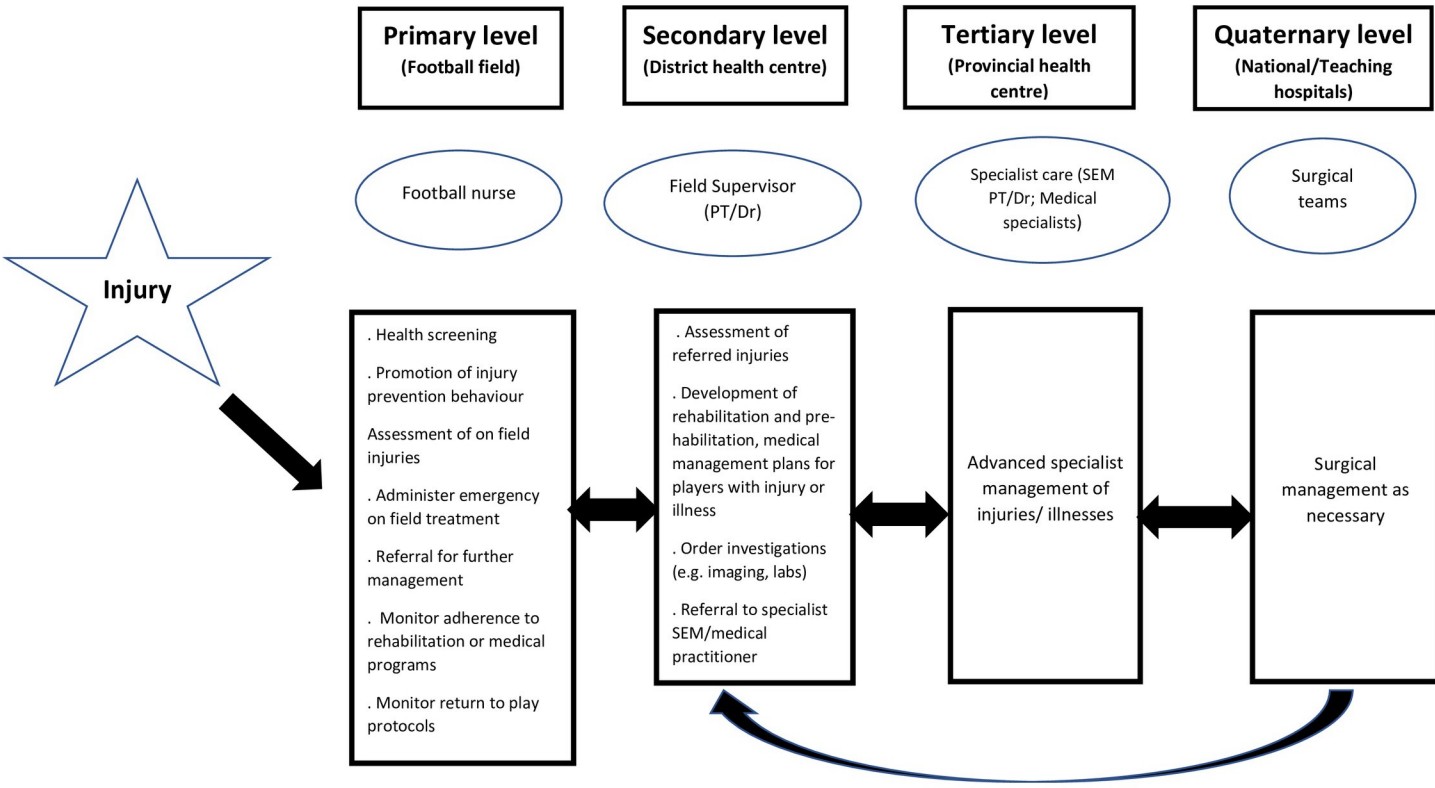

**Fig 1. Proposed referral pathways for players with injuries or illnesses during the FIFA football nurse intervention.**

## Trial design and study arms

**Trial design.** This study will be a cluster randomised controlled trial. This design is applicable as it allows for a field-based intervention to be implemented for one competitive season while decreasing the chances of contamination between the groups. African women's football is the sport that has been chosen as it is the most commonly played women's team sport on the continent and therefore, is a priority sport for improving medical care at and from grassroots levels. Adherence to the FIFA Football Nurse program; football exposure data; injury, illness and medication use surveillance and the in this trial will be similar to those used in previous football studies [35, 37] and conform with the definitions and data collection procedures outlined in the latest injury consensus statements [36]. The study will also include a qualitative evaluation of the participants' (players, coaches, nurses) experience of the FIFA football nurse program.

All the 24 women's teams in the Malawi Women's Super league will be randomly assigned in a 1:1 ratio to either the Football Nurse intervention (INT–12 teams) or the control group/ standard care (CON–12 teams) through a random draw. For the allocation schedule for random assignment to each arm, the names of the 24 teams will be placed in three (3) bowls, a bowl for each region. They will then be picked consecutively and alternately, and allocated into the INT and CON groups by one of the investigators (NSM). This will be done after the training of the nurses and after each team (cluster) have agreed to be a part of the study. For transparency, the random draw will be conducted in the presence of representatives from the football association and will be video and audio- recorded.

**Control or standard care.** In the control or standard care group (CON) consisting of 12 teams (clusters), each team will have 30 registered players, one (1) team manager, and one (1) coach and will continue their SEM practise 'as per usual'. A member of the research team will contact a designated member of their technical personnel to collect data on football exposure; injury, illness and medication use surveillance after each training session and league match. Beyond this, the teams in the CON group will not receive any special instructions. It is noteworthy that the teams in the CON and INT groups will at some point in the season play against each other and the Football Nurse will be duty bound to render care should a player from the CON group be injured or fall ill during the match. Therefore, contamination cannot be entirely avoided.

**Football nurse intervention.** In the intervention group consisting of 12 teams (clusters), each team will have 30 registered players, one (1) team manager, and one (1) coach. Each team in the intervention group will be have one (1) trained general nurse assigned to them for training and match cover for the duration of one competitive season. The nurses will be supervised by and directly report to a physiotherapist or medical doctor to whom they will refer serious injuries, investigations and further management/rehabilitation as needed.

**Sample size.** The study aims to evaluate the impact of the Football nurse intervention on team sports medicine practice. Hence, the study will recruit twelve (12) nurses, with one assigned to each intervention team, and six (6) physiotherapists and/or doctors to supervise the Football Nurses from two teams each.

To determine the appropriate sample size, the study used the cluster randomized formula as in previous studies [38]. This formula considers the intra-cluster correlation coefficient, number of participants, expected effect size, and power of the study. The formula is as follows:

$$C = 1 + \left( Z_{\alpha/2} + Z_{\beta} \right)^2 \frac{\left[ \sigma_0^2 + \sigma_1^2 \right] \times \left[ 1 + (m - 1)k \right]}{m(\mu_0 - \mu_1)^2}$$

Where C is the number of clusters required per treatment arm, $Z_{\alpha/2}$ and $Z_\beta$ are the critical values of the standard normal distribution for the desired confidence level and power, respectively, $m$ is the number of individuals per cluster and $k$ is the intracluster coefficient, $\mu_1$ and $\mu_0$ are the true means, and $\sigma_1$ and $\sigma_0$ are the standard deviations, of the outcome variable in the presence and absence of the intervention, respectively.

The design effect of a CRT is related to the intracluster correlation coefficient as follows:

$$DEFF = 1 + (m - 1)k$$

Assuming a 95% confidence level ($Z_{\alpha/2} = 1.96$), a power of 80% ($Z_\beta = 0.84$), an expected effect size of 0.19 (based on a 50% increase in team sports medicine practice), a standard deviation of 0.30, an intra-cluster correlation coefficient of $k = 0.2$, and 25 participants per cluster, the study estimates that it will need 12 clusters (teams) in both the control and intervention groups. Therefore, a total of 600 participants will be recruited, with 300 participants in each group. In summary, the study will recruit twelve (12) nurses and six (6) physiotherapists/doctors to supervise the intervention teams. The study will need 12 clusters (teams) in both the control and intervention groups, with a total of 600 participants (300 in each group) to detect a statistically significant change in team sports medicine practice at a 95% confidence level and 80% power, using the cluster randomized formula [38, 39].

**Recruitment.** The 24 teams that will be targeted for this intervention compete in the Malawi Super League, which is the highest organised league for women's football in Malawi. The level of football is amateur, as is most women's football in LMICs. However, the league produces many of the players for the national teams. The teams in the league are divided into eight (8) teams each by geographical region into the Northern, Central and Southern regions. The distribution of teams into the two arms per region is shown in **Fig 2** below.

While medical resources required for football training and matches are known, most teams do not have medical support available to them either on a part-time or full-time basis. The 24 teams will be recruited and monitored for the 2022/2023 football season. The nurses will be approached through their District Health Officers (DHOs) while the teams will be approached through the country's football governing body; Football Association of Malawi (FAM) and their specific District Sport Officers (DSOs) as is regulated in Malawi. A presentation on the trial will be made to the various DHOs, DSOs, and FAM by one of the investigators (EC), after which representatives from the teams will be invited to attend a presentation on the study protocol, where they will be invited to participate in the study in either the INT or CON group. The inclusion and exclusion criteria for the trial are as shown in **Table 2**.

Once the team has confirmed their participation, all the players registered for the team in the 2022/2023 season will be eligible to participate and will be recruited through the team. Verbal and written assent (players under 18) and consent (in English or Chichewa) from the nurses, players, and technical personnel will be required for each participant before the they can be included in the trial as mandated by Malawian research ethics and government statutes.

**Blinding.** However, participants will not be aware of the entire 'true' purpose of the trial. They will be informed that they are being allocated a Football Nurse so that we can study the feasibility of this project in alleviating shortage of medical support in women's football. Likewise, the INT group cannot be blinded as they will be aware of the Football Nurse assigned to them. However, the data will be coded and blinded before it can be analysed statistically.

**Standardisation of procedures.** The physiotherapists and doctors will complete the modules from the freely and publicly available online FIFA Medical Diploma before the trial commences. This will allow for uniformity of football medicine practice and supervision among the participants during the trial. Shortly before the football season begins (1 week before), all

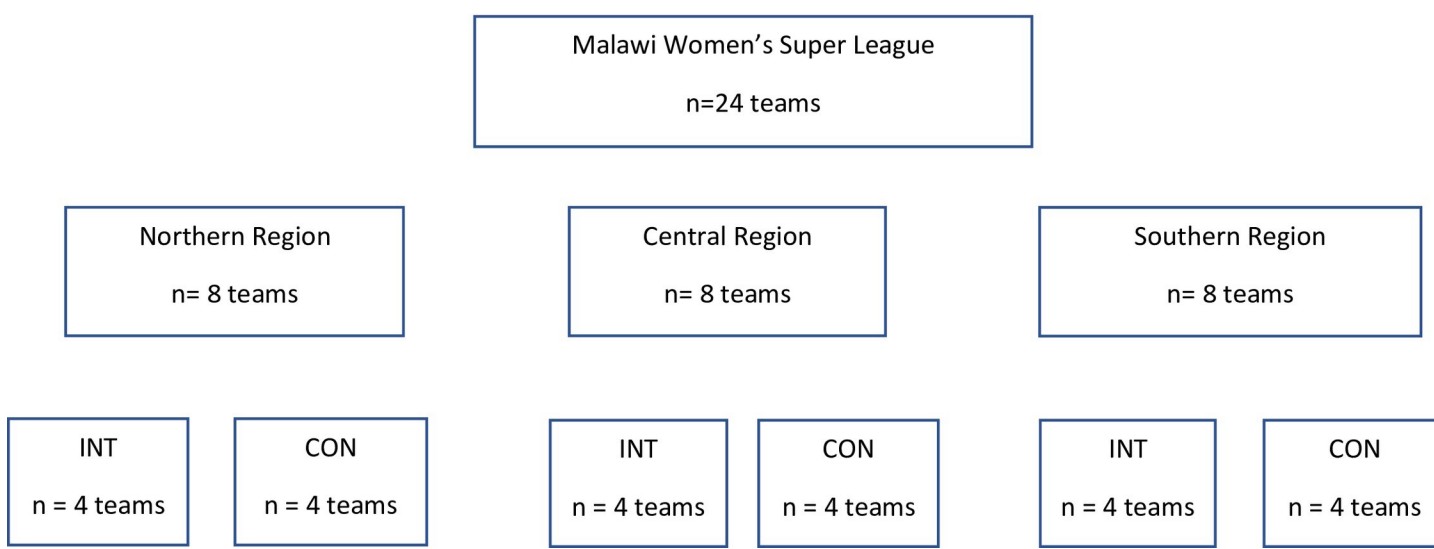

**Fig 2. FIFA football nurse recruitment strategy for participating teams in the Malawi Women's Super league.**

the nurses and the physiotherapists/doctors will undergo a one (1) day training on emergency pitch side football medicine using the latest version of the FIFA Emergency Care training manuals. This training will be conducted by experienced sports medicine personnel (NSM and EC) as well as experienced trainers in basic life support (BLS) and first aid from the Departments of Emergency and Orthopaedics at KUHeS. Further, all research personnel will receive half a day's training on the online data capturing system that will be used for the study. The research team will have access to written standard operating procedure on their mobile data collection devices as well. The team will also meet every fortnight to catch up and ensure adherence to standard research procedures.

**Table 2. Inclusion and exclusion criteria for nurses, women's football teams and physiotherapists/doctors for the FIFA football nurse trial.**

| Inclusion criteria | Exclusion criteria |
|---|---|
| Women's football clubs in the Women's League in Malawi registered with the Football Association of Malawi (FAM). | Women's football clubs in the Women's League in Malawi that already have medical personnel attached to their team. |
| Nurses registered with the Nurses and Midwives Council of Malawi with at least 3 years post graduate experience. | For potential supervisors, less than 3 years post qualification experience; incomplete FIFA Medical Diploma. |
| Physiotherapists with a recognised university qualification in physiotherapy, registered with the Medical and Dental Council of Malawi and with at least 3 years post graduate practice experience. | For potential Football Nurses, less than 3 years of post-qualification experience. |
| Medical doctors with a recognised university qualification (e.g., MBChB/MBBS) registered with the Medical and Dental Association of Malawi and at least 3 years post graduate practice experience. | For potential Football Nurses and supervisors, current attachment to a Women's Football club in Malawi's Women's League |
| (for physiotherapists and medical doctors) Completion of the online FIFA Medical Diploma prior to the commencement of the project. | |

The football nurse attached to each team will be primarily responsible for data collection for the INT teams and the research team, through a designated individual in the football team, will be responsible for data collection for the CON teams.

## Outcome measures

The schedule of enrolment and assessments that will be conducted in the study is shown in **Fig 3**.

**Primary outcome measure.** *Current team medical practice*. Sports medicine practice will be evaluated at a cluster (team) level. This will be a self-administered questionnaire, which evaluates the teams' current medical practice, their players and coaches' sources of medical information, and their self-reported knowledge of injury management. It is adapted from previous work on medical practice in African sporting federations [11].

**Secondary outcome measures.** *Demographic data and past medical history*. This will be a self- administered questionnaire on age, level of education, injury, and medical history. Additionally, players will complete a section on their menstrual history; namely, age at menarche, use of contraception, obstetric history, menstrual hygiene management practices, presence and management of menstrual cycle disorders as well as perceived effect of the menstrual cycle on football performance. This outcome will be evaluated at an individual participant level.

*Injury prevention and management behaviours*. A self-administered questionnaire will be used to evaluate injury prevention and management behaviours at an individual participant level as reported in other work [18, 40]. These will explore players' and coaches' knowledge, beliefs, and practices of injury prevention in football.

*Injury risk*. Individual intrinsic and extrinsic Injury risk will be assessed. Intrinsic injury risk will be assessed using functional movement screening (FMS^TM), injury history, musculoskeletal assessment, and muscle endurance testing. The FMS consists of seven tests: overhead squat, single-leg hurdle, split squat, shoulder mobility, active straight leg raise, stability push-up and rotary stability. The musculoskeletal assessments will be; limb girths, ankle dorsiflexion (weight bearing lunge test), sit and reach, lumbar forward flexion, and lumbar spine extension. While muscle endurance testing will comprise prone elbow plank and bilateral side planks to exhaustion, and the number of push-ups and squats performed in 60 seconds. All these tests have been reported elsewhere [38] and can be performed with minimal equipment and were specifically chosen for their ease of administration by an individual even with minimal training. Extrinsic injury risk will include assessment of the team's home training field players' personal equipment and stadium health services as done by other researchers [11].

*Nutritional data*. Individual nutritional and dietary assessments will be performed using the 24-hour food recall questionnaires, which records one's meals in the last 24 hours, the main sources of the food, timing of meals and the food portions [41].

*Mental health assessment*. This will be assessed using the Sports Mental Health Assessment Screening Tool-1 (SMART-1). The SMART-1 is a standardised assessment tool that aims to identify individual athletes potentially at risk for or already experiencing mental health symptoms and disorders to facilitate timely referral for those in need of support or treatment [42].

*Football exposure*: *Injury, illness, and medication use surveillance*. These data will be collected at a cluster level using injury, illness and football load forms as recommended in injury consensus statements [36]. The research team will follow up with each football nurse on a weekly basis to ensure surveillance data is being captured. Should an injury occur that requires investigations or further management, the nurses will refer to the PT/Dr, who will proceed with the treatment, design/prescribe further management and document accordingly. League performance will also be collated from FAM at the end of the season as this has previously been associated with injury rates in other team sports [43].

| | STUDY PERIOD | | | | |
|---|---|---|---|---|---|
| | Enrolment | Allocation | Post-allocation | | Close-out |
| | | | T1 | T2 | T3 |
| TIMEPOINT | Pre-study | 0 | Baseline (season start) | Mid-season | End of season |
| **ENROLMENT:** | | | | | |
| **Eligibility screen** | X | | | | |
| **Informed consent** | X | | | | |
| **Training nurses/ physiotherapists** | X | | | | |
| **Allocation of teams** | | X | | | |
| **INTERVENTIONS:** | | | | | |
| **ASSESSMENTS:** | | | | | |
| Demographic data and past medical history questionnaire | | | X | | |
| Injury risk assessment | | | X | X | X |
| Injury prevention behaviours questionnaire | | | X | X | X |
| Team medical practice questionnaire | | | X | X | X |
| 24h food recall questionnaire | | | X | X | X |
| Sports Mental Health Assessment Screening Tool-1 (SMART-1) | | | X | X | X |
| Football exposure | | | ◆————————————————◆ | | |
| Injury surveillance report | | | ◆————————————————◆ | | |
| Illness surveillance report | | | ◆————————————————◆ | | |
| Medication use surveillance | | | ◆————————————————◆ | | |

**Fig 3. Schedule of enrolment and assessments for the FIFA football nurse protocol.**

Assessment of injury behaviours, team medical practice, injury risk, nutritional data, and mental health assessment will be done at three time points: at the beginning before the trial begins, halfway through the trial, and at the end of the season. Football exposure and surveillance data will be collected and collated at the end of each training session and/or match until the end of the season.

*Measurement of potential confounders.* Despite all players playing in the same league and at the same level, primary confounders for injury risk include physical/ anthropometric characteristics; existing mental health or psychiatric conditions and home environment, which may influence the SMHAT- 1 values, for example. Secondary confounders include playing experience, playing position, and playing level as well as socio-economic status, which will affect nutrition, menstrual hygiene management and private management of injuries. Where possible, these will be considered in the analyses. Some of them will also be explored further in the qualitative interviews at closure of the protocol.

*Adherence.* This study has been designed to be optimally implemented in grassroots women's football in a low-income setting. The researchers will monitor adherence of the nurses' attendance at training sessions and matches using electronic record sheets. These records will then allow for exposure hours to be calculated and a measure of adherence to be determined. The surveillance data will be monitored on a biweekly basis to ensure it is up to date and following standardised operating procedures. The study is being performed in a league that has massive shortages of medical personnel and therefore, the participants in the intervention group will be motivated to adhere to the protocol as best they can.

*Ethical considerations.* Permission was sought and obtained from FAM and Directors of District and Social Services (DHSS) in all the regions where the women football teams are based. Ethical approval for this trial was obtained from the College of Medicine Research and Ethics Committee (COMREC), Blantyre, Malawi (protocol number: P.01/22/3551). The study will be conducted in accordance with the Declaration of Helsinki, the protocol and GCP guidelines. Before they can be a part of the study, and before randomisation, the investigators will explain the objectives of the study to the physiotherapists, doctors and nurses; football teams, their women players, and their coaching teams. They will only then be eligible to be included in the randomisation and eventual participation in the study after they have provided verbal and written assent and consent. This study will be voluntary, and participants will be informed that they have a right to participate or to withdraw from the study at any time, without prejudice. Data collected from the study will be stored in a password protected database and computer and that will only be accessible to the researchers. Study participants will be assured of their confidentiality. The names of the participants will only be on the consent form and in all other instances, they will be referred to by their assigned unique code.

In the event of a participant being flagged as being at risk of self-harm or harm to others in the mental health screening, the Football Nurses and their supervisors will be obliged to provide brief intervention within the scope of their practice, following which they will refer the participant for further management by the relevant healthcare providers. Should a medical condition be detected in the physical assessments, a similar approach will be taken in the management of the player.

*Data analysis.* Data will be analyzed using the IBM Statistical Package for Social Sciences (SPSS) version 21. The Shapiro-Wilk test for normality of the data will be conducted. Baseline comparability of participants for potential confounding factors such as age, length of football career, and baseline injury risk will be conducted using a chi-square test (for categorical variables) and either independent t-tests [normally distributed data] or Mann-Whitney U tests [non-normally distributed data] (for continuous variables).

Due to the relatively small number of clusters, data analyses will be based on summary level measures to account for the shortcomings of individual level regression methods. The primary variable of interest will be sports medicine practice. The mean (SD) sports medicine practice score will be calculated and presented by cluster and by arm. The mean of these mean scores and their associated 95% confidence intervals (CI) will then be presented for each arm. Linear regression of the mean sports medicine practice scores per cluster and per arm; two-way

analysis of variance (ANOVA), will be used to explore the differences in sports medicine practice score and the 95% CI associated with the FIFA Football Nurse intervention by arm.

For the injury, illness, and medication use data, injury, illness, or medication use history will be recorded to determine the baseline injury, illness, or medication use risk. Injury, illness, and medication use rates and corresponding 95% CIs will be calculated for all players as the number of injuries/illnesses/medications reported per 1000 hours of football exposure (training and match). The severity of the injuries/illnesses, as scored by the number of days of missed training or matches, will also be calculated. Additional analysis will include, but not be limited to, the incidence and type of injury/illness/medication; chronic versus acute injuries/illnesses/medications; contact versus non-contact injuries and mechanism of injury (all per 1000 player hours). A Poisson regression will be used to compare differences in the injury, illness, or medication use rates between the control and intervention arms. To evaluate the impact of the FIFA Football nurse intervention on injury/illness/medication use outcomes, we will use logistic regression analysis to compare the proportion of players with injuries/illness/taking medications between the intervention and control arms, adjusting for potential confounding variables such as previous injury, age, playing experience, BMI, and playing conditions. We will also conduct subgroup analyses to determine whether the intervention has differential effects on different subgroups of players.

For the testing data such as injury risk (FMS, musculoskeletal screening), nutritional data (24h recall), and mental health assessment (SMHART-1), a mixed between-within-subjects ANOVA will compare the two arms over the whole intervention, then a post hoc Bonferroni test will be used to determine whether there are significant changes in the various scores and at what time point during the intervention these occur. Should there be a significant change in test scores, a one-way ANOVA will be performed at the specific time point to determine which specific test is different between arms. Additionally, a mixed effects logistic regression model will be used to examine the effects of arm assignment (intervention vs control) and time for categorical outcomes (e.g., sports injury prevention behaviours). To assess adherence and feasibility, descriptive statistics will be used to determine which aspects of the FIFA Football Nurse programme are best or worst implemented. Sensitivity analyses include adjustment for baseline team sports medicine practice and other potential confounding factors (e.g., previous injury, age, football experience, BMI, socioeconomic status). We will also include tests for effect-modification by relevant factors.

Additionally, data from the semi-structured interviews will be transcribed verbatim and a thematic analysis will be conducted using Braun and Clarke's 6 step model [44] to group common themes and describe the experiences of women's football stakeholders, and the barriers and facilitators towards the football nurse task-sharing approach. Analysis will be approached inductively by tagging meaning units which will then listed as themes, compared, and clustered to form categories, then assigned as general dimensions. Inter-rater reliability checks will be conducted by a blinded researcher, who will code a subset of the manuscripts and determine congruence of resulting themes. If clustering effects are observed in the data, these will be accounted for using appropriate statistical methods such as multilevel or hierarchical models.

All statistical tests conducted in this trial will be two-sided and p-values of $< 0.05$ will be considered statistically significant.

*Patient and public involvement.* Patients and the public were not involved in the design of this protocol. However, various stakeholders in Malawi women's football (e.g., nurses, women football players, district sports officers, district representatives in women's football) will be consulted on the conduct, reporting, and dissemination plans for the research.

*Project time frame*. This project will be performed over an 8-month period during the 2022/2023 football season (May to December).

- May 2022 –training of nurses; recruitment of football teams; randomisation of teams.

- June 2022 –baseline testing of teams (assessment of injury behaviours, team medical practice, injury risk, nutritional data, and mental health assessment)

- June 2022 to December 2022 –implementation of the football nurse intervention; injury surveillance; adherence and exposure data collection.

- October 2022 –midway testing of teams (assessment of injury behaviours, team medical practice, injury risk, nutritional data, and mental health assessment)

- December 2022 –final testing of teams (assessment of injury behaviours, team medical practice, injury risk, nutritional data, and mental health assessment); semi-structured interviews with sample of players and coaches from both groups and all nurses

*Outcomes and significance*. We expect to develop a low cost, sustainable and context relevant solution to manage the treatment gap of football injuries/illnesses in underserved communities such as women's football. Through the study, we also anticipate the development and maintenance of player health, injury and illness monitoring databases for Malawian women football players in club football, which can be replicated in grassroots women's football leagues in LMICs. This will allow the monitoring of trends and development of relevant preventive or management strategies in these settings. Additionally, we expect this study to lead to the development of a flagship model which can inform implementation of similar task sharing approaches to SEM practice in other LMICs in Africa and globally.

## Supporting information

**S1 Checklist. SPIRIT-outcomes 2022 checklist (for combined completion of SPIRIT 2013 and SPIRIT-outcomes 2022 items)[a].**
(PDF)

**S1 Protocol. Ethics committee approved protocol version.**
(PDF)

## Acknowledgments

The authors acknowledge the support of Solomon Mudege, Salome Iiyambo, and Arijana Demirovic from FIFA and Gomezgani Zakazaka, Alfred Gunda, and Walter Manda from the Football Association of Malawi.

## Author Contributions

**Conceptualization:** Nonhlanhla Sharon Mkumbuzi.

**Data curation:** Ben Sorowen.

**Formal analysis:** Ben Sorowen.

**Funding acquisition:** Nonhlanhla Sharon Mkumbuzi, Enock Madalitso Chisati.

**Investigation:** Nonhlanhla Sharon Mkumbuzi, Andrew Massey, Samuel Kiwanuka Lubega, Enock Madalitso Chisati.

**Methodology:** Andrew Massey, Samuel Kiwanuka Lubega, Ben Sorowen,
Enock Madalitso Chisati.

**Project administration:** Enock Madalitso Chisati.

**Supervision:** Andrew Massey, Enock Madalitso Chisati.

**Writing – original draft:** Nonhlanhla Sharon Mkumbuzi.

**Writing – review & editing:** Andrew Massey, Samuel Kiwanuka Lubega,
Enock Madalitso Chisati.

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
