## [Decision Letter · Decision Letter 0]

5 Dec 2022

PONE-D-22-30766FIFA Football Nurse – A task sharing approach in Sports and Exercise Medicine practice in grassroots women’s football in low- and middle- income settings. A study protocol for a parallel randomised controlled trialPLOS ONE

Dear Dr. *Mkumbuzi*,

Thank you for submitting your manuscript to PLOS ONE. After careful consideration, we feel that it has merit but does not fully meet PLOS ONE’s publication criteria as it currently stands. Therefore, we invite you to submit a revised version of the manuscript that addresses the points raised during the review process. Please, address the reviewer comment as well as the Editor comment as appended. 

We look forward to receiving your revised manuscript.

Kind regards,

Samuel Bosomprah

Academic Editor

PLOS ONE

Journal Requirements:

"The FIFA Football Nurse project is funded by the Women’s Football Development Department at Federation Internationale Football Association (FIFA). The funders were not involved in the design or write up of the protocol."

"We have read the journal's policy and the authors of this manuscript have the following competing interests: AM is Director of the Medical Department at FIFA; NSM is a technical consultant for FIFA’s Women’s Football Department."

7. We note that the original protocol file you uploaded contains a confidentiality notice indicating that the protocol may not be shared publicly or be published. Please note, however, that the PLOS Editorial Policy requires that the original protocol be published alongside your manuscript in the event of acceptance. Please note that should your paper be accepted, all content including the protocol will be published under the Creative Commons Attribution (CC BY) 4.0 license, which means that it will be freely available online, and any third party is permitted to access, download, copy, distribute, and use these materials in any way, even commercially, with proper attribution.

Therefore, we ask that you please seek permission from the study sponsor or body imposing the restriction on sharing this document to publish this protocol under CC BY 4.0 if your work is accepted. We kindly ask that you upload a formal statement signed by an institutional representative clarifying whether you will be able to comply with this policy. Additionally, please upload a clean copy of the protocol with the confidentiality notice (and any copyrighted institutional logos or signatures) removed.

Additional Editor Comments:

a)The protocol proposes to randomise 12 football teams to receive an intervention and 12 to receive a control.

b) The teams are made up of individuals and therefore this should be described as a Custer randomised controlled trial.

c) The protocol should be guided by the "Consort 2010 statement: extension to cluster randomised trials".

d) The authors stated 6 objectives. But it is recommended to identify one as the primary objective and all others as secondary.

e) I recommend that the Objective 6 should be rephrased as the primary objective, which must have hypothesis to operationalised the objective by stating the primary outcome and its measurement scale.

f) The current protocol has too many outcomes:

1. Increased knowledge of and practice of injury prevention strategies;

2. Reduced intrinsic and extrinsic injury risk

3. Lower incidence and prevalence of injuries, illnesses, and medication use

4. Increased referral of injuries to other healthcare providers

5. Increased completion of injury rehabilitation protocols.

I recommend for the authors to identify one primary outcome and use the rest as secondary and power the study on the primary outcome noting that this is a cluster RCT. If the authors decided to keep the 5 outcomes, this has implication on the design of the study including sample size calculation and analysis to adjust for multiple primary outcomes and indicate what a success looks like (effect on at least one outcome or effect on all outcomes????).

g) Revise the statistical analysis plan to reflect the guidance I have provided here. Use statistical methods appropriate for analysing cluster randomised trials.

h) Revise the title to include cluster randomised trial.

Reviewers' comments:

Reviewer's Responses to Questions

**Comments to the Author**

1. Does the manuscript provide a valid rationale for the proposed study, with clearly identified and justified research questions?

Reviewer #1: Yes

2. Is the protocol technically sound and planned in a manner that will lead to a meaningful outcome and allow testing the stated hypotheses?

Reviewer #1: Partly

3. Is the methodology feasible and described in sufficient detail to allow the work to be replicable?

Reviewer #1: No

4. Have the authors described where all data underlying the findings will be made available when the study is complete?

Reviewer #1: No

5. Is the manuscript presented in an intelligible fashion and written in standard English?

Reviewer #1: Yes

6. Review Comments to the Author

You may also provide optional suggestions and comments to authors that they might find helpful in planning their study.

Reviewer #1: In this study protocol, a two-arm cluster randomized control trial is being proposed to evaluate the effectiveness of implementing the intervention of Football Nurses for Women’s Football League in Malawi. The primary aim is to train FIFA Football Nurses. The secondary aim is to evaluate the effect of trained nurses on sports injury related outcomes compared to controls. The feasibility of training nurses will also be evaluated.

Minor revisions:

1- The specific objectives on pages 16 and 17 do not correspond to the objectives identified in the abstract. Additionally, the specific objectives fail to mention cost and sustainability.

2- State that the study will be a cluster randomized controlled study.

3- Page 20: Ratio is misspelled.

4- Sample size section: To improve clarity, replace “attached” with “assigned.”

5- Sample size justification: Include comprehensive sample size/power justification by stating the statistical testing method and the power of the test.

6- Data Analysis section: State how categorical variables will be summarized. Typically this includes frequencies and percentages.

7- Consider revising this statement because it is not accurate. “Should there be a statistically significant change at a specific time point, then a one-way ANOVA will be performed to determine the specific outcome that is different between the groups.” Consider the following: “Should there be a statistically significant overall time by group effect, step-down tests will be performed to determine at which time points the two groups differ.”

8-To assist in the review process, add line numbering to the document.

7. PLOS authors have the option to publish the peer review history of their article (what does this mean?). If published, this will include your full peer review and any attached files.

Reviewer #1: No

---

## [Author Response · Author response to Decision Letter 0]

6 Feb 2023

We have provided a comprehensive response to the reviewers' comments in a separate document, "Response to Reviewers"

---

## [Editor Report · Decision Letter 1]

13 Feb 2023

PONE-D-22-30766R1FIFA Football Nurse – A task sharing approach in Sports and Exercise Medicine practice in grassroots women’s football in low- and middle- income settings.  A study protocol for a cluster randomised control trialPLOS ONE

Dear Dr. *Mkumbuzi,*

Thank you for submitting your manuscript to PLOS ONE. After careful consideration, we feel that it has merit but does not fully meet PLOS ONE’s publication criteria as it currently stands. Therefore, we invite you to submit a revised version of the manuscript that addresses the points raised during the review process. The Editor's comment is as appended below.

We look forward to receiving your revised manuscript.

Kind regards,

Samuel Bosomprah

Academic Editor

PLOS ONE

Additional Editor Comments:

• There are several outcomes in the current primary objective including: sports medicine practices, injury prevention behaviours; injury risk parameters; incidence and prevalence of injuries and illnesses in teams.

• The sample size calculation was based on “sports medicine practices”, suggesting that sports medicine practices is the primary outcome of interest. Therefore, authors should revise the primary objective to read as follows:

Primary Objective:

To estimate the effect of football nurse intervention on sports medicine practices during one competitive season in Malawi’s Women’s football league. Hypothesis: Compared to teams without a Football Nurse, the [mean??] sports medicine practices will be higher among teams with a Football Nurse during one competitive season in Malawi’s Women’s football league.

• The authors defined “sports medicine practices” as a mean and yet they used sample size formular for proportion? Can the authors use appropriate sample size formular.

• Authors should add the remaining components (minus sport medicine practice) of the current primary objective to the list of secondary objectives as: Compare injury prevention behaviours; injury risk parameters; incidence and prevalence of injuries and illnesses in teams with and without a Football Nurse during one competitive season in Malawi’s Women’s football league.

• The statistical analysis plan looks like a shopping list of statistical techniques as if they don’t know which techniques suits what? For each outcome (or group of outcomes measured on similar scale), the authors should indicate the technique to use. For example, the authors should decide on whether linear regression or ANOVA is appropriate for cluster level analysis of continuous outcomes; decide on “t-tests or Mann-Whitney U tests, for continuous variables”???? etc

• The authors are encouraged to read “Cluster Randomised Trials by Richard J. Hayes and Lawrence H. Moulten” for insight into analysis of cluster RCT.

• The subtitle in line 294 should read “Control or Standard of Care”

• The subtitle in line 306 should read “Football nurse intervention”

• The authors should integrate the “RCT methodology” section into the “ Trial design” section to read logically.
---

## [Author Response · Author response to Decision Letter 1]

8 Mar 2023

Please refer to the attached document "Response to Reviewers".

---

## [Editor Report · Decision Letter 2]

18 Apr 2023

FIFA Football Nurse – A task sharing approach in Sports and Exercise Medicine practice in grassroots women’s football in low- and middle- income settings.  A study protocol for a cluster randomised control trial

PONE-D-22-30766R2

Dear **Dr. Mkumbuzi**,

We’re pleased to inform you that your manuscript has been judged scientifically suitable for publication and will be formally accepted for publication once it meets all outstanding technical requirements.

Kind regards,

Samuel Bosomprah

Academic Editor

PLOS ONE
---

## [Editor Report · Acceptance letter]

24 Apr 2023

PONE-D-22-30766R2 

FIFA Football Nurse – A task sharing approach in Sports and Exercise Medicine practice in grassroots women’s football in low- and middle- income settings. A study protocol for a cluster randomised controlled trial 

Dear Dr. Mkumbuzi:

I'm pleased to inform you that your manuscript has been deemed suitable for publication in PLOS ONE. Congratulations! Your manuscript is now with our production department. 

Kind regards, 

on behalf of

Dr. Samuel Bosomprah 

Academic Editor

PLOS ONE